# Effect of Soluble Dietary Fiber of Navel Orange Peel Prepared by Mixed Solid-State Fermentation on the Quality of Jelly

**DOI:** 10.3390/foods12081724

**Published:** 2023-04-20

**Authors:** Yanan Cheng, Puyou Xue, Yi Chen, Jianhua Xie, Guanyi Peng, Shenglan Tian, Xinxin Chang, Qiang Yu

**Affiliations:** State Key Laboratory of Food Science and Technology, China-Canada Joint Laboratory of Food Science and Technology (Nanchang), Key Laboratory of Bioactive Polysaccharides of Jiangxi Province, Nanchang University, 235 Nanjing East Road, Nanchang 330047, China; yanan01005@163.com (Y.C.);

**Keywords:** navel orange peel, soluble dietary fiber, mixed solid-state fermentation, structural and functional properties, jelly

## Abstract

The aim of this work was to prepare soluble dietary fibers (SDFs) from insoluble dietary fiber of navel orange peel (NOP-IDF) by mixed solid-state fermentation (M-SDF) and to investigate the influence of fermentation modification on the structural and functional characteristics of SDF in comparison with untreated soluble dietary fiber (U-SDF) of NOP-IDF. Based on this, the contribution of two kinds of SDF to the texture and microstructure of jelly was further examined. The analysis of scanning electron microscopy indicated that M-SDF exhibited a loose structure. The analysis of scanning electron microscopy indicated that M-SDF exhibited a loose structure. In addition, M-SDF exhibited increased molecular weight and elevated thermal stability, and had significantly higher relative crystallinity than U-SDF. Fermentation modified the monosaccharide composition and ratio of SDF, as compared to U-SDF. The above results pointed out that the mixed solid-state fermentation contributed to alteration of the SDF structure. Furthermore, the water holding capacity and oil holding capacity of M-SDF were 5.68 ± 0.36 g/g and 5.04 ± 0.04 g/g, which were about six times and two times of U-SDF, respectively. Notably, the cholesterol adsorption capacity of M-SDF was highest at pH 7.0 (12.88 ± 0.15 g/g) and simultaneously exhibited better glucose adsorption capacity. In addition, jellies containing M-SDF exhibited a higher hardness of 751.15 than U-SDF, as well as better gumminess and chewiness. At the same time, the jelly added with M-SDF performed a homogeneous porous mesh structure, which contributed to keeping the texture of the jelly. In general, M-SDF displayed much excellent structural and functional properties, which could be utilized to develop functional food.

## 1. Introduction

Gannan navel orange, a representative citrus, is regarded as one of the renowned specialty products in Jiangxi Province, China, which contain numerous beneficial phytochemicals [1]. It has been acknowledged that orange peel provided a variety of active substances such as dietary fiber, flavonoids, alkaloids, essential oils, pectin, and carotenoids, which play a critical role in antioxidant, hypoglycemic, antitumor, and other health effects [2]. Navel orange peel (NOP), as a primary by-product, possesses excellent nutritional value, but, generally, is an industrial waste to be dumped in the process [3]. The accumulation or improper management of the peels would lead to microbial fermentation and putrefaction, which could pose serious environmental and ecological issues. However, NOP is an important source of natural ingredients, particularly dietary fiber (DF), which might be utilized to create beneficial substitutes [4].

Dietary fiber (DF) is widely considered to be the seventh nutrient and has been a research hotspot in functional foods and other fields. Depending on the solubility, DF is defined as soluble dietary fiber (SDF) and insoluble dietary fiber (IDF), respectively. SDF has better functional activity, higher viscosity, and a larger potential for gel formation than IDF, which has great application value in the food industry [5,6]. In addition, studies have shown that SDF could effectively diminish blood glucose and plasma cholesterol, attenuate cardiovascular diseases, and regulate the composition of intestinal bacteria, which is conducive to human health [7,8]. However, IDF is predominant (60–80%) in most dietary fiber from natural plants. In recent years, more and more researchers have adopted modification methods to rise the yield of SDF [9]. Considering these, exploring an efficient modification method to obtain SDF from IDF is a promising strategy.

The modification methods of DF commonly include physical, enzymatic, and chemical methods [10], while these methods exhibit low conversion efficiency. Microbial fermentation technique is an efficient and green modified method for preparing SDF. Moreover, the functional properties of SDF obtained by fermentation, such as water-holding capacity (WHC) and swelling capacity, were considerably improved [11]. In fact, mixed solid-state fermentation (SSF), as a natural and economical method, is an effective technique to enhance the content of nutritional substances from plants [12]. Our previous study has confirmed the same result; however, enzymes produced by single strains suffer from an incomplete enzyme system and low single-enzyme activity. In contrast, mixed SSF can enhance the ratio of enzyme composition and overall enzyme activity, which have attracted more attention from researchers. Ahamed et al. significantly increased the production of cellulase systems using mixed SSF via *Trichoderma reesei* (*T. reesei*) and *Aspergillus Niger* (*A. niger*) [13].

Jelly is identified as a healthy food and is mainly composed of starch, water, sugar, carrageenan, konjac flour, etc. In the traditional processing, starch is poorly soluble, heterogeneous, and unstable in jelly preparation. On the contrary, SDF enables us to fill up these weaknesses since it not only improves the viscosity and WHC of jelly, but also empowers jelly to regulate the intestinal tract and improve constipation [14,15].

We have demonstrated in our previous studies that fermentation was able to convert IDF to SDF and enhance its functional activity. However, to our knowledge, there was less information about the preparation of SDF from IDF in navel orange peel by mixed solid-state fermentation. Therefore, this study evaluated the differences in structural and functional properties of SDFs converted by mixed SSF (named M-SDF) and untreated (named U-SDF) to explore the impact of mixed solid-state fermentation modification on SDF. In addition, they were added to estimate the effect of both SDFs on the quality of jelly and their application value. This would provide a theoretical basis for the development of jelly with beneficial properties, as well as support the preparation and application of high-quality functional dietary fiber products at the technical level.

## 2. Materials and Methods

### 2.1. Experiment Materials

Gannan navel orange (Citrus sinensis Osbeck cv. Newhall), fresh, disease-free, and pests-free, was supplied with Chongyi County of Ganzhou City, China, in December 2019. The peels were dried for 36 h in a constant temperature blast drying oven at 55 °C, then crushed and passed through a 100-US mesh to obtain NOP powder. The NOP-IDF was derived using the enzymatic methodology [16]. *Trichoderma reesei* (GDMCC 3.537) and *Aspergillus niger* (GDMCC 3.576) were bought from the Microbial Culture Collection Center (Guangdong, China). Heat-stable α-amylase (200,000 U/g) was obtained from Aladdin Biotechnology and its enzyme activity was determined by DNS method. Papain (50,000 U/g) was bought out of Pangbo Biological Engineering Co., Ltd. (Naning, China). and its enzyme activity was measured by using tyrosine as a substrate. All other chemical reagents were of analytical purity.

### 2.2. Preparation of U-SDF

U-SDF was extracted by the enzymatic extraction described by Zhang’s method [16], with slight modifications. The navel orange peel powder was first mixed with deionized water (1:10, *w*/*v*) and then incubated with 1% α-amylase for 30 min at 66 °C in a water bath, in order to remove the starch. The mixture was then cooled to 60 °C, with the pH adjusted to 6.0 and 0.5% papain was added to incubate for 60 min to remove the protein. The resulting mixture was centrifuged for 15 min at 4800 r/min, and the soluble fiber enriched supernatant was washed with 95% ethanol (1:4 *v*/*v*) and distilled water, lyophilized, and stored in the refrigerator at −20 °C.

### 2.3. Preparation of M-SDF

M-SDF was extracted from NOP-IDF by mixed SSF with *T. reesei* and *A. niger*. The mixed SSF condition was on the basis of our prior research: 3:1 inoculation ratio of *T. reesei* to *A. niger*, delayed inoculation time of 28 h for *A. niger*, and an initial pH of 6.5. The following procedure for extracting M-SDF was the same with the one in Section 2.2.

### 2.4. Structure Characteristics

#### 2.4.1. Scanning Electron Microscopy (SEM)

U-SDF and M-SDF surface morphology were separately evaluated by scanning electron microscopy (SEM) (JSM 6701F, JEOL, Tokyo, Japan). The specific operation steps are referred to the method of Huang et al. with a slight modification [17]. The samples were placed on a sample stage and sprayed with gold for 60 s under a 5 kV accelerating voltage. The morphology and structure of the samples were observed and shot in enlargement multiple ×1000 and ×3000.

#### 2.4.2. Molecular Weight (Mw)

The molecular weight of U-SDF and M-SDF samples were analyzed by a Waters high-performance liquid chromatograph (HPLC) apparatus with reference to the method before [18]. The mobile phase was a saturated sodium chloride solution in the current study. The standard curve was produced using the Dextran T system standard (Mw: 10 kDa, 25 kDa, 40 kDa, 500 kDa) and glucose. All solutions were filtered through a 0.22 μm membrane filter and then injected into a chromatographic system. The Mw of each sample was estimated according to the obtained standard curve.

#### 2.4.3. X-ray Diffraction (XRD)

The X-ray diffractometer (D8 Advance, Bruker, Saarbrucken, Germany) was used to get the XRD profiles of the U-SDF and M-SDF samples. The operating voltage and current were set at 30 kV and 20 mA, respectively, and the diffraction angular range was scanned from 2° to 60° (2*θ*). The crystallinity indices (Ic) were calculated with the Segal’s method [19].

#### 2.4.4. Fourier Transfer-Infrared Spectrometry (FT-IR)

The characteristic functional groups of SDF were determined with reference to the method of Yang et al. [20]. The FT-IR spectrums of the freeze-dried samples were scanned 32 times in the range from 4000 cm^−1^ to 400 cm^−1^ using the Nicolet iS50 spectrometer (Thermo Fisher Scientific, Waltham, MA, USA) with a resolution of 4 cm^−1^. The spectrums were measured after each sample was gently covered on the surface of the ATR crystal.

#### 2.4.5. Thermal Properties

The thermal properties of U-SDF and M-SDF were performed using thermal gravimetric analysis (TGA), as described previously with slight variations [21]. The parameters of thermos gravimetric analyzer (TGA 4000, PerkinElmer, Waltham, MA, USA) were set as follows: temperature range 30–600 °C, heating rate 20 °C/min.

#### 2.4.6. Monosaccharide Composition

The monosaccharide compositions of U-SDF and M-SDF were determined by high-performance anion exchange chromatography (HPAEC) (Dionex ICS-5000, Dionex Corporation, Sunnyvale, CA, USA) coupled with a pulse amperometric detection (PAD) [22]. In brief, each sample (5 mg) was mixed in 0.5 mL H_2_SO_4_ (12 M) in an ice bath and stirred using a magnetic stirrer in a plug test tube for 0.5 h until completely dissolved. Afterwards, 2.5 mL ultra-pure water was added to the mixture and kept stirring in an oil bath for 4 h at 105 °C. After cooling, the reaction solution was diluted to a 50 mL volumetric flask, at which point the concentration was 100 μg/mL. The solution was filtered through a 0.22 μm membrane filter before being injected into the ion chromatography system.

### 2.5. Functional Properties

#### 2.5.1. Water Solubility (WS)

1 g SDF sample was mixed with 50 mL water, followed by stirring in a water bath at 90 °C for 20 min. After centrifugation, the supernatant was lyophilized and weighed. The results were defined using the following Equation (1):WS (g/g) = W_2_/W_1_(1)
where W_1_ is the weight of SDF and W_2_ is the weight of supernatant after freeze-drying.

#### 2.5.2. Water Holding Capacity (WHC)

0.5 g lyophilized SDF sample was dissolved in 10 mL ultra-pure water, and incubated at 37 °C for 2 h. The precipitate was obtained after centrifugation (4800× *g*,10 min), and then weighed immediately. WHC was expressed using the following Equation (2):WHC (g/g) = (M_2_ − M_1_)/M_1_(2)
where M_1_ is the weight of SDF sample and M_2_ is the weight of precipitate after centrifugation.

#### 2.5.3. Oil Holding Capacity (OHC)

The difference in the determination of OHC was that the solvent was soybean oil compared with Section 2.5.2. OHC was expressed using the following Equation (3):OHC (g/g) = (M_2_ − M_1_)/M_1_(3)
where M_1_ is the weight of SDF sample and M_2_ is the weight of the precipitate after oil absorption.

#### 2.5.4. Cholesterol Adsorption Capacity (CAC)

CAC of U-SDF and M-SDF samples were evaluated using the method reported by Jia et al., with a few modifications [11]. The egg yolk was thoroughly mixed with distilled water (1:9, *v*/*v*) to make an emulsion. The SDF sample (0.1 g) was added to 5 mL emulsion with pH adjusted to 2.0 and 7.0 to simulate both the stomach and small intestine environment of humans, then the mixture was periodically shaken for 2 h in a water bath at 37 °C. After centrifugation at 4800× *g* for 10 min, the supernatant was diluted 10 times with glacial acetic acid and the cholesterol content was determined. SDF-free emulsion was set as a blank. CAC was calculated using the following Equation (4):CAC (mg/g) = (C_1_ − C_2_)/W_1_(4)
where C_1_ was the weight of cholesterol in the blank group, C_2_ was the weight of cholesterol in the sample adsorption solution, and W_1_ was the weight of the SDF sample.

#### 2.5.5. Glucose Adsorption Capacity (GAC)

GAC was determined by the method with slightly modified [23]. Briefly, each SDF sample was mixed with 0.5 mg/mL of glucose solution and kept for 2 h at 37 °C water bath before centrifugation. The supernatant (0.5 mL), ultra-pure water (2.5 mL), and dinitro-salicylate (DNS) color development reagent (2 mL) were mixed to form a reaction system, which was incubated for 6 min at 100 °C in a water bath. When the reaction solution returned to room temperature, the glucose concentration of the supernatant was measured at 520 nm. GAC was calculated using the following Equation (5):GAC (mg/g) = (G_1_ − G_2_)/W_1_(5)
where G_1_ and G_2_ was the weight of glucose in the solution before and after adsorption, W_1_ was the weight of SDF sample.

### 2.6. Dietary Fiber Jelly

#### 2.6.1. Jelly-Making Procedure

At first, the mixture of 15% sugar, 0.5% konjac gum, 0.5% carrageenan gum, 81% water, and 3% SDF (U-SDF and M-SDF) was added sequentially to the beaker and then the combination was continuously agitated and allowed to boil at 100 °C for 30 min. Subsequently, the beaker was removed and 0.18% citric acid was immediately added. Finally, the residue was filled into the preprepared mold. The jelly was stored in the refrigerator at 4 °C. All samples were cylinders of the same volume (3.0 cm in diameter and 2.0 cm in height).

#### 2.6.2. Texture Profile Analysis (TPA)

The TPA of dietary fiber jelly was determined by a texture profile analyzer (TA-XT plus, Stable Co., London, UK) equipped with P/50R probe [24]. The operating parameters were set as follows: the TPA secondary compression mode, all test speeds were 2 mm/s, 50% test strain, trigger force was 5.0 g, and the trigger type was automatic.

#### 2.6.3. Scanning Electron Microscopy of Jelly

Evaluation of the apparent morphology of freeze-dried samples of jelly added with U-SDF and M-SDF respectively by SEM (JSM 6701F, JEOL, Tokyo, Japan). The morphology and structure of the samples were observed under the magnification of ×300 and ×1000 under the acceleration voltage of 10 kV.

### 2.7. Statistical Analysis

All experimental conditions were repeated in triplicate, and results were presented as mean ± standard deviation. Statistical significance of the data was analyzed by Duncan’s test while performed with IBM SPSS Statistics 21 statistical package.

## 3. Results and Discussion

### 3.1. Structure Analysis

#### 3.1.1. Morphological Observation

Scanning electron microscopy is one of the most important means to explore the microstructure of DF, which was utilized to compare the microstructure features of the U-SDF and M-SDF in this study. Comparing two micrographs of the same multiple from Figure 1, the morphology and structure of U-SDF and M-SDF were evidently different. Specifically, the whole structure of M-SDF was sparse and porous, which appeared in a network-like structure and possessed a large specific surface area. On the contrary, it was distinctly observed that the U-SDF was relatively smooth and consisted of intact large fibrous shells. The morphological changes perhaps were owing to the removal of more loose protein and starch around SDF by mixed SSF [24]. The loose structure facilitates molecules to be immobilized in the network structure or pores, which possibly promotes the hydration and adsorption properties of samples.

#### 3.1.2. M_w_ Measurement

To the best of our knowledge, M_w_ is a significant parameter for assessing the functional characteristics of SDF. In addition, M_w_ showed a negative correlation with solubility but a positive relationship with cohesion and viscosity. Therefore, HPGPC was applied to evaluate the differences of molecular weight between U-SDF and M-SDF, and their average M_w_ data were calculated based on the calibration equations derived from the linear regression of calibration curves as shown in Figure 2. It was apparent that the elution times of the main elution peaks were similar for U-SDF and M-SDF with 16.3, 17.9, and 18.9 min, respectively. Remarkably, four sharp main peaks were presented in the elution curve of M-SDF (Figure 2B), whereas there were only three peaks in U-SDF. In particular, we found a new fraction emerging in M-SDF and verified that the NOP-IDF transformation was attributed to the hybrid SSF, which was consistent with our previous study [11]. A possible explanation for this might be that microbial action destroyed the structure of dietary fiber, thus transforming and producing new molecular fragments during the fermentation process.

#### 3.1.3. XRD Analysis

In order to study the crystal structure changes of SDF treated by mixed SSF, XRD analysis was carried out (Figure 3). The XRD patterns of the two samples were similar, with a characteristic wide diffraction peak at 23.4° 2*θ* for M-SDF and a characteristic diffraction peak at 22.3° 2*θ* for U-SDF. It showed that the two SDF samples were mainly amorphous structures [19]. Furthermore, the M-SDF sample possessed a new characteristic crystalline peak at 12.5° 2*θ*, indicating a significant change in the crystal structure of M-SDF after mixed SSF. By computing, the crystallinity of M-SDF (31.25%) was significantly higher than that of U-SDF (24.76%), which could be the degradation of crystal structure materials such as hemicellulose during mixed SSF, resulting in a more regular molecular arrangement and thus increasing the relative proportion of crystalline cellulose. In addition, changes in the crystal structure of M-SDF might affect its thermal stability and other functional properties.

#### 3.1.4. FT-IR Analysis

The functional groups of U-SDF and M-SDF were determined using FT-IR spectroscopy. As shown in Figure 4, the spectral profiles of the two samples were similar in characteristics. Specifically, the broad and strong peak near 3268.8 cm^−1^ may be caused by the -OH bond vibration, which was related to the presence of water and hydrogen bonds in SDF [25]. There were weak absorption bonds in the vicinity of 2931.3 cm^−1^ and 1737.5 cm^−1^, suggesting that some methylene was presented in SDF samples. The absorption peaks near 1596.8 cm^−1^ attributed to the stretching of C=O bond, which suggested that both U-SDF and M-SDF samples contained uronic acid. The most interesting aspect of this graph was that the intensity of the C=O bond peak decreased dramatically in M-SDF compared to U-SDF, indicating a higher uronic acid content of the sample because of reconfiguration. The absorption peak at 1409.7 cm^−1^ is attributed to the C-H vibration of the aromatic lignin moiety. The absorption peak near 1010.5 cm^−1^ might be caused by ether bond stretching [26]. Additionally, the peak of M-SDF near 952.7 cm^−1^ suggested that the stretching vibration induced by β-glycosidic bond was enhanced after modification. Overall, the FT-IR spectra of U-SDF and M-SDF both showed characteristic absorption peaks of polysaccharides, and the functional groups of the two samples were similar. These results suggested that hydrophilic and other active groups in SDF would not change by mixed SSF.

#### 3.1.5. Thermal Properties

Thermal analysis could be employed to examine the thermal stability and composition of substances, which was regarded as an essential structural characteristic of SDF. The TGA curves of U-SDF and M-SDF were shown in Figure 5. From the picture, it could be seen that the thermal degradation of SDF included four stages: drying, pre-carbonation, carbonation, and combustion [11]. The temperature of the drying phase for SDF pyrolysis was below 150 °C. At this stage, the water absorbed by U-SDF and M-SDF was evaporated, which led to a slight decline in the weight of the sample. Among them, the faster mass loss of M-SDF indicated its comparatively positive water absorption [27]. When the temperature reached the range of 150–230 °C, it was the pre-carbonation stage, a slow process of the thermal decomposition of macromolecules into small molecules. As can be seen from Figure 5, the mass loss of M-SDF was minimal compared with U-SDF. Then the third carbonation stage was at the range of 230–300 °C. The weight was rapidly lost and the polysaccharide was thermally decomposed, which was because of the breakage of hydrogen bonds in the molecules and the degradation of SDF. According to the thermal analysis curve, it was obvious that the mass of M-SDF decreased severely. The temperature of the combustion stage was above 300 °C, the SDF residue was completely burned. Finally, the residual masses of U-SDF and M-SDF were 29.43% and 39.69%, respectively.

In summary, the results showed that the weight loss of M-SDF was visibly lower than that of U-SDF, which might be correlated with its higher crystallinity index, in agreement with the discussion in the XRD analysis [28]. The M-SDF obtained by mixed SSF emerged with active thermal stability, which was more valuable than U-SDF for most food processing and production.

#### 3.1.6. Monosaccharide Composition Analysis

The monosaccharide composition results of U-SDF and M-SDF were presented in Table 1. The xylose existed in the U-SDF sample was defined as 1, and other monosaccharides were compared with it. Eight monosaccharides were detected in M-SDF, including rhamnose, arabinose, galactose, glucose, xylose, mannose, galacturonic acid, and glucuronic acid. However, rhamnose was not found in U-SDF and fructose contained in U-SDF was not detected in M-SDF. This might be attributed to part of glycosidic bonds in NOP-IDF that were broken down by mixed SSF, and hemicellulose was converted to M-SDF during the process. It was worth noting that the molar ratio of galactose was significantly higher in M-SDF than in U-SDF. Closer inspection of the table showed M-SDF contained more galacturonic acid and glucuronic acid, which was consistent with the FTIR analysis results. The lower glucose content in M-SDF could be explained by the fact that a large amount of glucose was consumed during mixed SSF. The above results indicated that the monosaccharide compositions of SDF before and after mixed SSF were not only changed, but also the ratios of monosaccharides were rearranged.

### 3.2. Functional Property Analysis

#### 3.2.1. WS, WHC, and OHC

WS was defined as the property of a substance to form an aqueous solution, which was an important reference index for SDF. The results showed that the WS of M-SDF was significantly higher than U-SDF (Table 2). According to the discussion of Section 3.1.1, the M-SDF presented a looser structure and thus exposed more hydrophilic groups, which made M-SDF have a better solubility. WHC was related to the viscosity of DF. The ability of SDF to connect with other compounds was enhanced by its high water-holding capacity, which helped prevent food shrinkage [29]. OHC contributed to stabilizing high-fat foods and acted as an emulsifier, a crucial reference indicator for SDF [30]. Table 2 showed that M-SDF was about six times WHC and two times OHC of U-SDF. The possible explanation for these significant differences might be that M-SDF provided a larger specific surface area generated numerous and exposed groups. As a result, water or oil could easily impregnate the DF and bond tightly to them for reducing losses [31].

#### 3.2.2. CAC Analysis

It was believed that SDF possessed the capacity to regulate several metabolic pathways of cholesterol in the body. Previous studies have demonstrated that SDF was able to reduce blood cholesterol levels and attenuate the risk of cardiovascular disease [32]. The results of both U-SDF and M-SDF (Table 2) showed that CAC at pH = 7.0 was greater than at pH = 2.0. Overall, SDF reflected higher CAC values under simulated intestinal conditions (pH = 7.0). Xu et al. had proved that SDF could inhibit the absorption of cholesterol in the small intestine and facilitate the rapid excretion of cholesterol [33]. In addition, it was noteworthy that M-SDF had significantly higher CAC than U-SDF at the same pH, which may be due to the more complex structure and larger contact area of M-SDF, thus allowing more cholesterol to be bound. In short, M-SDF exhibited greater potential for application in balancing cholesterol levels.

#### 3.2.3. GAC Analysis

Excessive intake of glucose would up-regulate the body’s blood glucose level, consequently leading to fat accumulation and inducing diabetes, fatty liver, and other diseases in turn. SDF could effectively down-regulated blood glucose levels in the body as it inhibited the binding of glucose in the digestive tract [34]. As seen in Table 2, M-SDF exhibited stronger GAC (*p* < 0.05), which might be owing to the different structures of U-SDF and M-SDF. This result may be explained by the fact that the mixed bacteria solid state fermentation treatment endowed SDF with a more loose and porous structure as well as exposed active groups efficiently, which could combine with glucose molecules on a larger scale and play a role in reducing blood sugar levels [35].

### 3.3. Jelly with U-SDF or M-SDF

#### 3.3.1. Texture of Jelly

The texture characteristics of jelly played a significant part in its product quality. As shown in Table 3, jelly with M-SDF had higher hardness, gumminess, and chewiness. The hardness of the jelly was determined by the swelling force of gelatinized starch particles [36]. The higher hardness of the jelly was obtained with the addition of M-SDF (2431.84 ± 236.39), it exhibited a hardness that was 751.15 higher than that added with U-SDF. It was likely associated with the higher crystallinity promoting the formation of a better network structure in the jelly. Related research has demonstrated that the gumminess was inversely correlated with amylose content in the starch gel. The jelly containing M-SDF constituted a better gel network structure, thus improving its gumminess [37]. Chewiness was measured for simulation by pressing the samples twice [38]. The results indicated that the jelly containing M-SDF had better chewiness (392.91 ± 198.44) and texture. Cohesiveness was the amount of cohesion required to form a jelly sample, while springiness and resilience refer to the ability of a jelly sample to maintain its integrity and resist damage. The springiness, cohesiveness, and resilience of jelly containing M-SDF were lower than those containing U-SDF, but the difference was not significant and had little effect on jelly products.

#### 3.3.2. SEM of Jelly

Scanning electron micrographs (SEM) were used to observe the surface morphology of jelly. As shown in Figure 6, The jelly with the addition of M-SDF and U-SDF presented a different physical form. Specifically, the surface of M-SDF jelly was smooth and continuous with uniform organization, presenting a dense spatial structure, implying that the structure of jelly added with M-SDF was more stable. In addition, the porous structure means that the jelly possessed better adsorption capacity. It has been demonstrated that dietary fiber improved the structural properties of products such as yogurt and meat products by increasing viscosity, gel stability, and water-holding capacity [39,40]. Based on these, it was hypothesized that the particles of M-SDF might be embedded in the gel network of the jelly, making the microstructure tighter and more continuous. In contrast, the jelly containing U-SDF was not homogeneous and had a slight surface protrusion. This discrepancy would probably be attributed to the higher crystallinity of M-SDF, which enabled the molecules to be closely aligned, thus facilitating the formation of a better gel network structure in the jelly [41,42]. From the results, it can be inferred that the jelly containing M-SDF could enrich water effectively, maintain jelly texture, and appear better texture and chewiness, which further confirmed the conclusion in Section 3.3.1.

## 4. Conclusions

In conclusion, we discussed the differences in structural and functional properties between M-SDF modified from NOP-IDF by mixed SSF and U-SDF without treatment, and further investigated their effects on jelly properties. In contrast to U-SDF, M-SDF exhibited a looser structure, higher molecular weight and crystallinity, and more active thermal stability and monosaccharide compositions. In the aspect of functionality, M-SDF presented greater WS and bound water, oil, cholesterol, and glucose more efficiently. However, FT-IR results suggested that mixed SSF would not change the active groups in SDF samples. Furthermore, the addition of M-SDF could significantly improve the properties of jelly including hardness, gumminess, and chewiness. Compared with the jelly added with U-SDF, the surface of the jelly containing M-SDF was smooth and continuous, showing a uniform porous mesh structure. Therefore, M-SDF exhibited more excellent structural and functional properties, which could be utilized to develop functional products for food applications. Additionally, the present work might provide insights into the comprehensive utilization of fruit and vegetable by-products.

## Figures and Tables

**Figure 1 foods-12-01724-f001:**
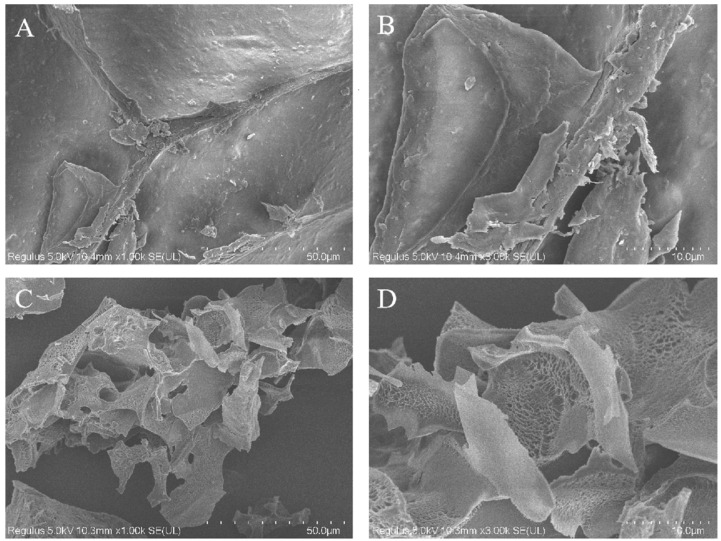
SEM micrographs of untreated soluble dietary fiber (U-SDF) from NOP ((**A**): ×1000 magnification; (**B**): ×3000 magnification) and soluble dietary fiber from NOP-IDF by mixed SSF (M-SDF) ((**C**): ×1000 magnification; (**D**): ×3000 magnification).

**Figure 2 foods-12-01724-f002:**
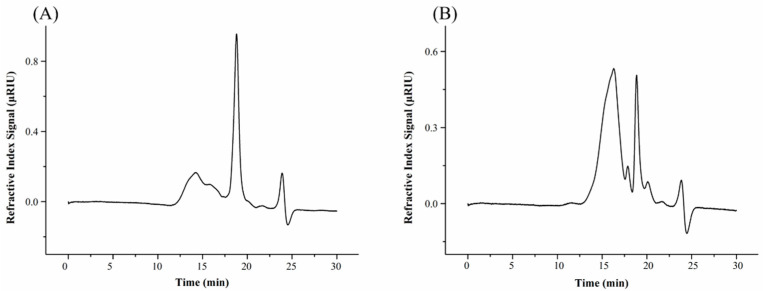
Gel permeation chromatogram profiles of U-SDF (**A**), M-SDF (**B**).

**Figure 3 foods-12-01724-f003:**
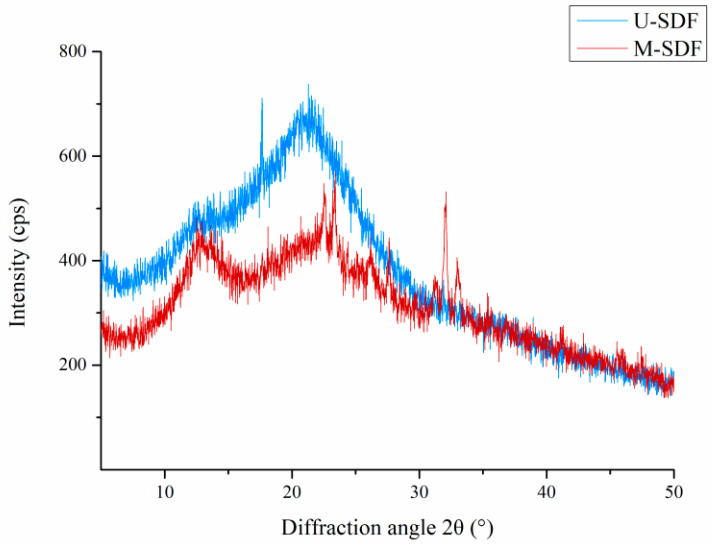
XRD pattern of the U-SDF and M-SDF.

**Figure 4 foods-12-01724-f004:**
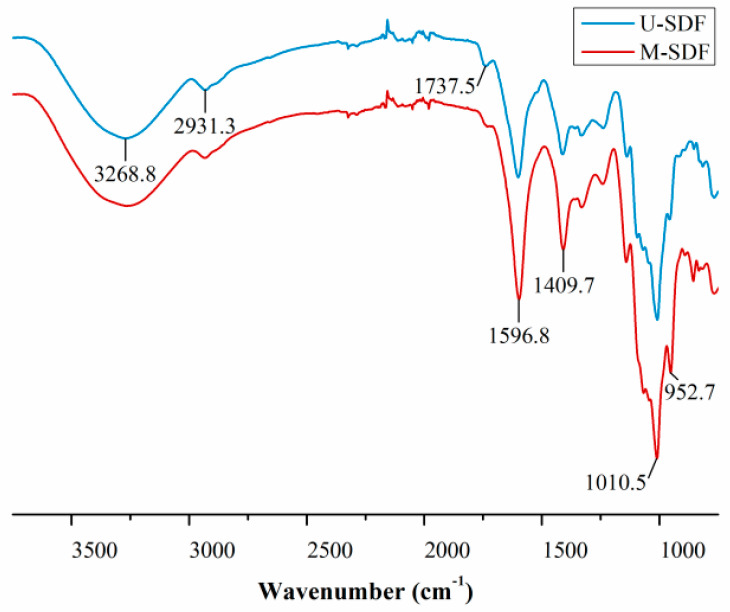
FT-IR spectra of the U-SDF and M-SDF.

**Figure 5 foods-12-01724-f005:**
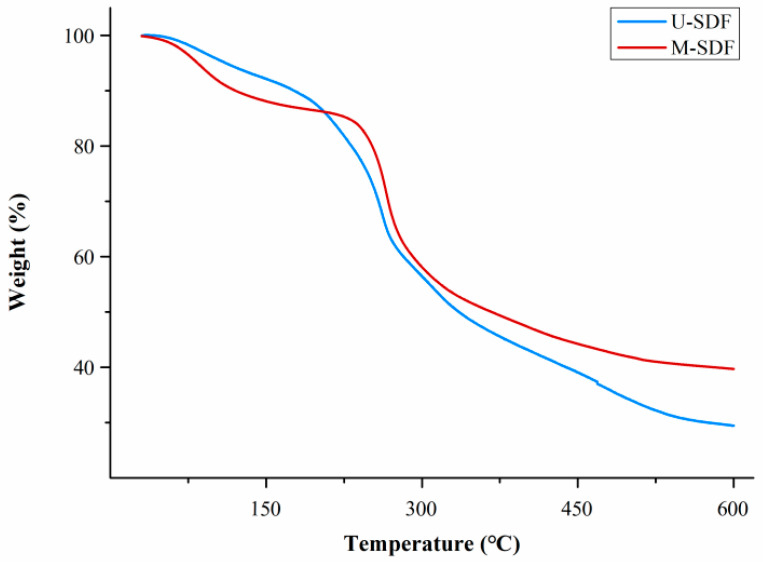
Thermal properties of the U-SDF and M-SDF.

**Figure 6 foods-12-01724-f006:**
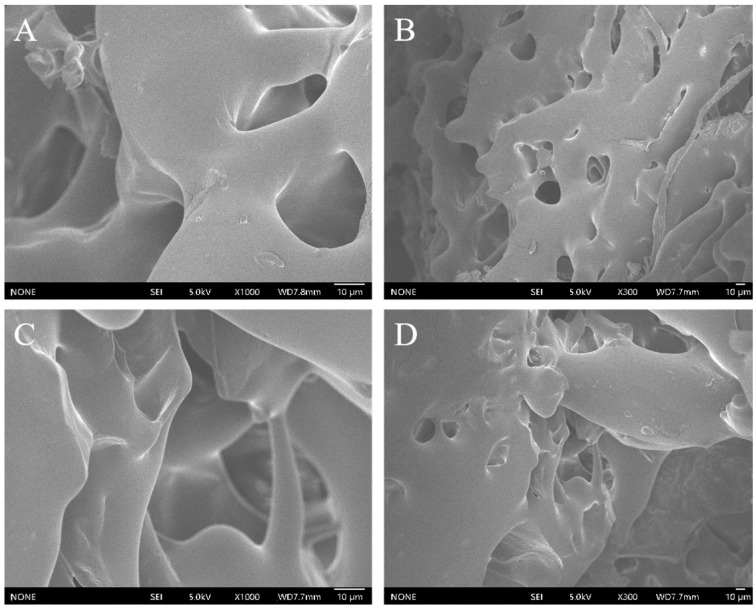
SEM micrographs of jelly added U-SDF ((**A**): ×1000 magnification; (**B**): ×300 magnification and jelly added M-SDF (**C**): ×1000 magnification; (**D**): ×300 magnification).

**Table 1 foods-12-01724-t001:** Monosaccharide composition ratio and relative mole % of samples.

Monosaccharide	U-SDF		M-SDF	
	Ratio	Mole %	Ratio	Mole %
Rhamnose	ND	0	3.35 ± 0.54	7.53 ± 0.91
Arabinose	11.31 ± 0.95	17.02 ± 0.45	7.53 ± 0.26	18.58 ± 0.92
Galactose	11.47 ± 0.97	14.38 ± 0.15	15.10 ± 0.71	31.04 ± 1.21
Glucose	35.22 ± 2.66	44.20 ± 0.50	2.01 ± 0.07	4.14 ± 0.31
Xylose	1.00 ± 0.00	1.51 ± 0.12	1.00 ± 0.00	2.47 ± 0.20
Mannose	2.45 ± 0.16	3.08 ± 0.08	2.68 ± 0.33	5.52 ± 0.78
Fructose	5.78 ± 2.54	7.39 ± 3.63	ND	0
Galacturonic acid	10.83 ± 4.24	12.41 ± 3.93	15.89 ± 2.89	30.11 ± 3.20
Glucuronic acid	ND	0	0.32 ± 0.06	0.60 ± 0.07

ND: not detected. Ratio refers to the ratio of different monosaccharides in different samples to xylose in the SDF sample. Mole % refers to the relative mole % of different monosaccharides in the same sample.

**Table 2 foods-12-01724-t002:** The effect of fermentation on the SDF functional properties.

Functional Properties		U-SDF	M-SDF
WS (g/g)		0.63 ± 0.02 ^b^	0.95 ± 0.04 ^a^
WHC (g/g)		1.00 ± 0.13 ^b^	5.68 ± 0.36 ^a^
OHC (g/g)		2.08 ± 0.03 ^b^	5.04 ± 0.04 ^a^
CAC (g/g)	pH = 2.0	4.68 ± 0.15 ^b^	9.62 ± 0.05 ^a^
	pH = 7.0	8.10 ± 0.14 ^b^	12.88 ± 0.15 ^a^
GAC (g/g)		7.94 ± 0.09 ^b^	10.42 ± 0.33 ^a^

Values represent mean ± standard deviations (*n* = 3). Significant differences (*p* < 0.05) in the same column were expressed using different letters (^a,b^).

**Table 3 foods-12-01724-t003:** Texture properties of jelly containing U-SDF or M-SDF.

Texture Properties	U-SDF	M-SDF
Hardness	1680.67 ± 170.94 ^b^	2431.84 ± 236.39 ^a^
Springiness	0.64 ± 0.05 ^a^	0.58 ± 0.04 ^b^
Cohesiveness	0.32 ± 0.10 ^a^	0.27 ± 0.11 ^ab^
Gumminess	554.39 ± 218.11 ^ab^	661.60 ± 308.4 ^a^
Chewiness	349.33 ± 131.29 ^ab^	392.91 ± 198.44 ^a^
Resilience	0.09 ± 0.03 ^a^	0.08 ± 0.04 ^ab^

Values represent mean ± standard deviations (*n* = 3). Significant differences (*p* < 0.05) in the same column were expressed using different letters (^a,b^).

## Data Availability

The data presented in this study are available on request from the corresponding author.

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
