# Peer review of "Effect of Soluble Dietary Fiber of Navel Orange Peel Prepared by Mixed Solid-State Fermentation on the Quality of Jelly"

_foods, 2023, doi:10.3390/foods12081724_

Round 1

Reviewer 1 Report

Comments:

In this paper, the results on the preparation on soluble dietary fibers (SDFs) from insoluble dietary fiber of orange peel (NOP-IDF) has been shown and discussed. Solid state fermentation was employed for this purpose. Beside it the structural analysis was performed using some structural methods. Manuscript is good and contributes significantly in the orange peel jelly formulation preparation. Authors are suggested to consider following comments to revise the manuscript.

-Title is not clear. “Navel” is a typo mistake. Navel orange peel???

- mixed solid-state fermentation (F-SDF) is a wrong abbreviation. It should be m-SDF.

-Abstract: It should have numerical findings. Currently, abstract is very plain.

-Orange peel characteristics in terms of compositional analysis should be added in Introduction.

-Section 2.1. Microorganisms name presentation should be in standard manner.

-Section 2.1. How were Papain and alpha-amylase activity were analysed? What were the methods employed? What is the meaning of 1 unit?

-What were the references or the methodology adopted for some examples such FTIR, SEM, Thermal properties of orange peel jelly?

-Figure 1. SEM results could be replaced with more clear SEM images.

-Table 1 characterization data should be presented in standard deviation.

-3.3.2.SEM of jelly should be elaborated with sufficient investigations documented in literature.

Author Response

  1. Title is not clear. “Navel” is a typo mistake. Navel orange peel???

Response 1: Sorry for our carelessness. We have corrected the misspelling of the word "Navel" in the title according to your advice.

  1. mixed solid-state fermentation (F-SDF) is a wrong abbreviation. It should be m-SDF.

Response 2: Thank you very much for your suggestion. In determining the abbreviation of the SDF samples, we primarily considered the importance of fermentation. However, your comments have greatly inspired us, and we agree that the abbreviation of “m-SDF” is essential to accept. In conclusion, we decided to change the abbreviation of mixed solid-state fermentation SDF to "M-SDF".

  1. Abstract: It should have numerical findings. Currently, abstract is very plain.

Response 3: Thank you very much for your suggestion. We have modified the abstract and added some specific numerical content. Please see the content in red in the Abstract section.

  1. Orange peel characteristics in terms of compositional analysis should be added in Introduction.

Response 4: Thank you very much for your suggestion. We have added a relevant description of the analysis of the nutritional components in orange peel and their corresponding functional activities. Please see paragraph 1 of the introduction.

  1. Section 2.1. Microorganisms name presentation should be in standard manner.

Response 5: Thank you very much for your suggestion. We have modified the name format of two microorganisms. Please see the red marked parts in section 2.1.

  1. Section 2.1. How were Papain and alpha-amylase activity were analysed? What were the methods employed? What is the meaning of 1 unit?

Response 6: Thank you very much for your questions. The amylase was bought from Aladdin Biotechnology Co., Ltd (Shanghai, China), and the target enzyme activity value provided in the manual was 200000 U/g. The target enzyme activity value of the papain purchased at Pangbo Bioengineering Co., Ltd respectively (Guizhou, China) was 50000 U/g. The units of amylase and papain was U/g according to the manual. We have checked the corresponding quality inspection certificate, and the results showed that the activity of the amylase and papain met the requirements. Moreover, the longest time interval between the use of amylase and papain was no more than six months, so the amylase and papain used could achieve the expected activity.

  1. What were the references or the methodology adopted for some examples such FTIR, SEM, Thermal properties of orange peel jelly?

Response 7: Thank you very much for your questions. The relevant references for FT-IR, SEM and thermal stability performance determination methods have been provided in the paper. Please see the red marked parts in section 2.4.1, 2.4.4 and 2.4.5.

  1. Figure 1. SEM results could be replaced with more clear SEM images.

Response 8: Thank you very much for your suggestion. We replaced a much clearer SEM microstructure image of the SDF. Please see the Figure 1.

  1. Table 1 characterization data should be presented in standard deviation.

Response 9: Thank you very much for your suggestion. We managed the data for the composition of monosaccharides in Table 1 and expressed them in the format of "mean ± standard deviation". Please see Table 1 in Section 3.1.6 for details.

  1. 3.2.SEM of jelly should be elaborated with sufficient investigations documented in literature.

Response 10: Thank you very much for your suggestion. We further consulted the relevant literature to provide supplementary illustrations of the SEM results of the jelly. Please see the description of the results in 3.3.2.

Reviewer 2 Report

The manuscript “Effect of soluble dietary fibre of navel peel prepared by mixed solid-state fermentation on the quality of jelly” deals with quite an exciting research theme with the partial transformation of IDF in SDF using a fermentation process. The only concern I found is the description of some of the methods employed in the study.

For example, in the 2.2 section. It is said that the NOP was incubated with geat-stable amylase. At what temperature? The 30 min treatment with amylase and 60 min with papain, where simultaneous or once one treatment was concluded, the other started?

Section 2.5.4- How do you know the cholesterol content in the egg yolk employed?

Author Response

  1. For example, in the 2.2 section. It is said that the NOP was incubated with geat-stable amylase. At what temperature? The 30 min treatment with amylase and 60 min with papain, where simultaneous or once one treatment was concluded, the other started?

Response 1: Thank you very much for your questions. The incubation of NOP with heat-stable α-amylase for starch removal was maintained at a water bath temperature of 66 °C.

The SDF extraction steps are as follows: The navel orange peel powder was first mixed with deionized water (1:10, w:v) and then incubated with 1% α-amylase for 30 min at 66 ℃ in a water bath, in order to remove the starch. After that, the mixture was cooled to 60 ℃, with the pH adjusted to 6.0 and 0.5% papain was added to incubate for 60 min to remove the protein. The resulting mixture was centrifuged for 15 min at 4800 r/min, and the soluble fiber enriched supernatant was washed with 95% ethanol (1:4 v/v) and distilled water, lyophilized and stored in the refrigerator at -20 ℃.

  1. Section 2.5.4- How do you know the cholesterol content in the egg yolk employed?

Response 2: Thank you very much for your questions. During the experiment, we measured a certain mass of egg yolk. The egg yolk and deionized water were mixed and homogenized into a homogeneous emulsion in the ratio of 1:9, and then SDF was added for adsorption experiments. Among them, the same uniform egg yolk emulsion was used for the blank and experimental groups to ensure the reliability of the experimental results.

Round 2

Reviewer 1 Report

I am satisfied with the authors changes in the revised manuscript. It can be accepted for publication.

Author Response

We would like to thank you for your efforts in reviewing our manuscript titled "Effect of soluble dietary fiber of Navel orange peel prepared by mixed solid-state fermentation on the quality of jelly", and providing many helpful commentsand suggestions, which will all prove invaluable in the revision andimprovement of our paper, as well as in quiding our research in thefuture.

We have studied your comments point by point, revised themanuscript accordingly. The amendments are highlighted in red in therevised manuscript. All authors have approved the response letter andthe revised version of the manuscript.

Thank you again for your valuable comments and suggestions.
